# Critical Appraisal of Large Vitamin D Randomized Controlled Trials

**DOI:** 10.3390/nu14020303

**Published:** 2022-01-12

**Authors:** Stefan Pilz, Christian Trummer, Verena Theiler-Schwetz, Martin R. Grübler, Nicolas D. Verheyen, Balazs Odler, Spyridon N. Karras, Armin Zittermann, Winfried März

**Affiliations:** 1Division of Endocrinology and Diabetology, Department of Internal Medicine, Medical University of Graz, 8036 Graz, Austria; christian.trummer@medunigraz.at (C.T.); verena.schwetz@medunigraz.at (V.T.-S.); martin.gruebler@gmx.net (M.R.G.); 2Division of Cardiology, Department of Internal Medicine, Medical University of Graz, 8036 Graz, Austria; nicolas.verheyen@medunigraz.at; 3Division of Nephrology, Department of Internal Medicine, Medical University of Graz, 8036 Graz, Austria; balazs.odler@medunigraz.at; 4National Scholarship Foundation, 55535 Thessaloniki, Greece; karraspiros@yahoo.gr; 5Clinic for Thoracic and Cardiovascular Surgery, Herz- und Diabeteszentrum Nordrhein-Westfalen (NRW), Ruhr University Bochum, 32545 Bad Oeynhausen, Germany; azittermann@hdz-nrw.de; 6Clinical Institute of Medical and Chemical Laboratory Diagnostics, Medical University of Graz, 8036 Graz, Austria; winfried.maerz@synlab.de; 7SYNLAB Academy, Synlab Holding Deutschland GmbH, 68159 Mannheim, Germany; 8V^th^ Department of Medicine (Nephrology, Hypertensiology, Rheumatology, Endocrinology, Diabetology, Lipidology), Medical Faculty Mannheim, University of Heidelberg, 68167 Mannheim, Germany

**Keywords:** vitamin D, RCT, clinical trial, cholecalciferol, randomized controlled trial, epidemiology, supplementation, mortality, infections

## Abstract

As a consequence of epidemiological studies showing significant associations of vitamin D deficiency with a variety of adverse extra-skeletal clinical outcomes including cardiovascular diseases, cancer, and mortality, large vitamin D randomized controlled trials (RCTs) have been designed and conducted over the last few years. The vast majority of these trials did not restrict their study populations to individuals with vitamin D deficiency, and some even allowed moderate vitamin D supplementation in the placebo groups. In these RCTs, there were no significant effects on the primary outcomes, including cancer, cardiovascular events, and mortality, but explorative outcome analyses and meta-analyses revealed indications for potential benefits such as reductions in cancer mortality or acute respiratory infections. Importantly, data from RCTs with relatively high doses of vitamin D supplementation did, by the vast majority, not show significant safety issues, except for trials in critically or severely ill patients or in those using very high intermittent vitamin D doses. The recent large vitamin D RCTs did not challenge the beneficial effects of vitamin D regarding rickets and osteomalacia, that therefore continue to provide the scientific basis for nutritional vitamin D guidelines and recommendations. There remains a great need to evaluate the effects of vitamin D treatment in populations with vitamin D deficiency or certain characteristics suggesting a high sensitivity to treatment. Outcomes and limitations of recently published large vitamin D RCTs must inform the design of future vitamin D or nutrition trials that should use more personalized approaches.

## 1. Introduction

Vitamin D deficiency is a public health problem because it has a relatively high worldwide prevalence and is associated with poor skeletal health, i.e., with rickets and osteomalacia [1,2,3]. Whether vitamin D exerts additional health benefits, in particular with reference to extra-skeletal diseases, is subject to an intense and ongoing debate within the scientific community.

Vitamin D has a unique metabolism, with its synthesis from endogenous precursors in the skin, a process induced by ultraviolet-B exposure, whereas dietary intake usually plays only a minor role [4]. Vitamin D is mainly metabolized to 25-hydroxyvitamin D (25(OH)D) in the liver and finally to 1,25-dihydroxyvitamin D (1,25(OH)2D, also termed calcitriol) in the kidneys. Importantly, 1,25(OH)2D is a steroid hormone. Its target receptors, i.e., vitamin D receptors (VDRs), are expressed in almost all human tissues [4]. In general, vitamin D can exert endocrine, paracrine, and autocrine effects as part of a complex regulation and interactions of vitamin D metabolism [4,5].

In mechanistic studies, VDR signaling has been implicated in the pathogenesis of many skeletal and extra-skeletal diseases [6]. In line with this, epidemiological studies have shown that serum concentrations of 25(OH)D, the accepted marker of vitamin D status, were inversely associated with adverse health outcomes including, e.g., cancer, cardiovascular diseases, and mortality, thus suggesting a widespread role of vitamin D for human health [6,7]. Based on this background, large randomized controlled trials (RCTs) were designed and have been recently published to evaluate the role of vitamin D supplementation for clinical outcomes with relevance for public health [8,9]. Cancer, cardiovascular and musculoskeletal health, infections, and mortality were the main outcomes of these trials and are thus the subject of this review. Given that the current opinion on vitamin D seems to be significantly, if not mainly, determined by these large RCTs, there exists a great need for a structured summary and guidance through all of these study results. While we acknowledge and appreciate previous work on this topic, we aim to extend these publications by considering important additional and modified aspects [2,10,11,12,13,14].

In this brief narrative review, we will summarize and discuss findings from major vitamin D RCTs and their potential impact on the appreciation of vitamin D regarding public health and science. We will start with a critical appraisal of clinical vitamin D research and then summarize major findings (primary outcomes) from large vitamin D RCTs that will then be discussed concerning their interpretations and implications. We will also outline current meta-analyses on vitamin D and major health outcomes before we provide an outlook on potential future research and clinical applications for vitamin D.

## 2. Critical Appraisal of Clinical Vitamin D Research

Scientific vitamin D publications have substantially increased over the last few decades, providing a wealth of data from experimental and observational studies. Overall, they support the hypothesis that vitamin D supplementation might be a sort of panacea for human health [15]. We agree that accumulating evidence points in the direction of a beneficial role of vitamin D in many diseases, thus justifying the call for large vitamin D RCTs. Nevertheless, before discussing the findings from these trials, a critical appraisal of vitamin D research leading to the initiation of these trials is warranted, as well as some basic considerations regarding general aspects of the trial designs.

Beforehand, we wish to stress the rather provocative but scientifically well supported statement that most claimed research findings are false, as pointed out by the landmark essay by Prof. John Ioannidis [16]. In summary, research findings are less likely to be true, the smaller the studies and the effect sizes are, the greater the number and the lesser the selection of tested relationships are, the greater the flexibility in designs, definitions, outcomes, and analytical models are, the greater the financial and other interests and prejudices are, and the hotter a scientific field (with more scientific teams involved) is [16]. While these considerations apply to clinical research in general, they definitely apply to the recent hype in vitamin D research and thus have to be taken into account when aiming for a balanced judgement of the scientific vitamin D literature.

As observational studies linking low 25(OH)D concentrations to poor health were one of the main drivers for the high public health interest in vitamin D, we want to underline that epidemiological studies on vitamin D status are particularly prone to confounding [17]. Low serum 25(OH)D concentrations are a consequence of, e.g., an unhealthy lifestyle with less outdoor activity (and thus less sun exposure), obesity, and poor nutrition [17]. In addition, reverse causation needs to be considered, which means that underlying diseases, in particular those related to inflammatory processes or limiting physical activity, may themselves decrease 25(OH)D concentrations [17]. Therefore, it has been postulated that a poor vitamin D status may merely be a marker of ill health [17]. These factors have been addressed in many well-conducted epidemiological vitamin D studies, but limitations inherent to observational study designs such as residual confounding or reverse causation cannot be completely excluded.

When designing large RCTs with the intention to prove or refute the hypothesis of a beneficial clinical value of vitamin D supplementation, it should be logical that findings from past nutrient RCTs are considered. The scientific literature contains several examples of vitamins (e.g., vitamin E) that showed promising results in experimental and association studies but that have failed to show benefits or that were even harmful in large RCTs [18,19,20]. When interpreting the results of these “disappointing” trials, a common conclusion was that large RCTs should primarily target populations who are particularly sensitive to the beneficial effects of the intervention, e.g., populations who are deficient for the respective vitamin. With reference to vitamin D, observational studies have largely shown that there is no meaningful association of 25(OH)D and health outcomes such as mortality over a wide range of the 25(OH)D distribution, whereas there was a steep increase in risk at very low 25(OH)D concentrations [21,22,23]. Unfortunately, large vitamin D RCTs were mainly designed to evaluate effects of a vitamin D supplementation in the general population with fairly “normal” 25(OH)D concentrations, thus not considering the findings from past nutrient RCTs mentioned above [8,9].

## 3. Results of Large Vitamin D RCTs

The main results, i.e., the primary outcomes, of three major vitamin D-randomized placebo-controlled trials in the general older population have been published recently: the Vitamin D and Omega-3 Trial (VITAL) from the United States, the Vitamin D Assessment (ViDA) study from New Zealand, and the Vitamin D3-Omega3-Home Exercise-Healthy Ageing and Longevity Trial (DO-HEALTH) from five different European countries (Switzerland, France, Germany, Portugal, and Austria) [24,25,26,27,28]. The characteristics and results of these trials are summarized in this chapter along with findings from a selection of recently published large vitamin D RCTs in specific populations that are considered relevant to current knowledge on vitamin D [29,30,31,32,33,34,35,36]. The main results of the D-Health Trial, an RCT with 21,315 participants aged 60 to 79 years old from Australia receiving either monthly doses of 60,000 IU of vitamin D or placebo are still pending [37]. The primary outcomes of this trial are all-cause mortality and total as well colorectal cancer incidence. Publications of a few other large vitamin D RCTs are also still pending [8,38]. All of the trials described below were randomized for vitamin D versus placebo in a 1:1 ratio if not otherwise specified, and effect sizes (e.g., hazard ratios (HRs)) are shown for the vitamin D group compared to the placebo group. When referring to vitamin D, we actually mean vitamin D3 (cholecalciferol) if not otherwise indicated.

### 3.1. VITAL

The VITAL study is a multicentre RCT from the United States among men aged 50 years or older and women aged 55 years or older with no history of cancer or cardiovascular diseases at baseline and who were (amongst other inclusion/exclusion criteria) required to agree to limit the use of vitamin D from all supplemental sources to 800 international units (IU) per day [27]. This trial was conducted using a two-by-two factorial design with 2000 international units (IU) of vitamin D per day and 1 g of marine n-3 fatty acids per day. Primary endpoints were invasive cancer of any type and major cardiovascular events (composite of myocardial infarction, stroke, or death from cardiovascular causes). A total of 25,871 participants were randomized (12,927 to vitamin D and 12,944 to placebo) and followed-up for a median of 5.3 years. Invasive cancer was diagnosed in 793 participants in the vitamin D group and in 824 participants in the placebo group, corresponding to a HR (with 95% confidence interval (CI)) of 0.96 (0.88 to 1.06; *p* = 0.47). Major cardiovascular events occurred in 396 participants from the vitamin D group and in 409 participants from the placebo group, translating into an HR (95% CI) of 0.97 (0.85 to 1.12; *p* = 0.69). At baseline, the current intake of vitamin D supplements (≤800 IU per day) was reported by 42.5% of the participants in the vitamin D group and by 42.7% of the participants in the placebo group. At 2 years, the prevalence of outside use of vitamin D supplements (>800 IU daily) was 3.8% in the vitamin D group and 5.6% in the placebo group, with an increase to 6.4% and 10.8%, respectively, after 5 years [27].

### 3.2. VIDA

The ViDA study is an RCT from New Zealand among adults aged 50 to 84 years old, with one of the exclusion criteria being the current use of vitamin D supplements, including cod liver oil at a dose of >600 IU per day if aged 50–70 years old and of >800 IU per day if aged 71–84 years old [25]. A total of 5108 participants received either vitamin D (n = 2558) at an initial dose of 200,000 IU followed by monthly doses of 100,000 IU one month later or a placebo (n = 2550) for a median of 3.3 years (range 2.5 to 4.2 years). Incident cardiovascular disease events, the primary outcome measure, were recorded in 303 participants from the vitamin D group and in 293 participants from the placebo group, resulting in an HR (95% CI) of 1.02 (0.87 to 1.29). At baseline, 8% of the study population were taking vitamin D supplements.

### 3.3. Do-Health

The DO-HEALTH study is a multicentre study from Europe in 2157 community dwelling adults aged 70 years or older with a 2 × 2 × 2 factorial design with three treatment comparisons, i.e., 2000 IU vitamin D per day, 1 g omega-3 fatty acids, and a strength-training exercise program [28]. The mean changes (with 99% CI) in the vitamin D versus the placebo group for the six primary outcomes were, −0.8 (−2.1 to 0.5; *p* = 0.13) mm Hg for systolic blood pressure, 0 (−0.7 to 0.8; *p* = 0.88) mm Hg for diastolic blood pressure, −0.1 (−0.3 to 0.1; *p* = 0.26) points for Short Physical Performance Battery, −0.1 (−0.4 to 0.1; *p* = 0.11) for the Montreal Cognitive Assessment (MoCA), 1.03 (0.75 to 1.43; *p* = 0.79) for nonvertebral fractures, and 0.95 (0.84 to 1.08; *p* = 0.33) for infections. At baseline, 10.8% of all of the study participants reported a supplement intake of 800 IU of vitamin D per day or more.

### 3.4. Vitamin D RCTs in Specific Populations

In the D2d study from the United States, 2423 participants with pre-diabetes were randomized to receive either vitamin D3 4000 IU per day or placebo [29]. After a median follow-up of 2.5 years, the HR (95% CI) for new onset diabetes was 0.88 (0.75 to 1.04; *p* = 0.12). In a RCT conducted in Bangladesh, 1300 pregnant women were randomly allocated into five groups to receive vitamin D supplementation either from 17 to 24 weeks of gestation until birth at a dose of 4200, 16,800, or 28,000 IU, respectively, per week, or from 17 to 24 weeks of gestation until 26 weeks postpartum at a dose of 28,000 IU peer week or placebo (equal group sizes) [30]. Among 1164 infants, at 1 year of age, there was no significant group difference in terms of the primary outcome, i.e., length-for-age z scores. In the Vitamin D to Improve Outcomes by Leveraging Early Treatment (VIOLET) trial, 1078 critically ill patients with 25(OH)D concentrations below 20 ng/mL (multiply by 2.496 to convert ng/mL to nmol/L) were randomized to receive a single enteral dose of 540,000 IU vitamin D or a placebo [31]. The 90-day mortality was 23.5% in the vitamin D and 20.6% in the placebo group, respectively, with a 2.9% (95% CI: −2.1 to 7.9%; *p* = 0.26) difference. Importantly, after the first interim analysis, the VIOLET trial was stopped for futility, i.e., the inability of this trial to achieve its objectives. An RCT from Nebraska (United States) randomized 2303 postmenopausal women aged 55 years old or older to either 2000 IU of vitamin D plus 1500 mg calcium per day or a placebo [32]. After four years, the HR (95% CI) for all-type cancer (excluding nonmelanoma skin cancer) was 0.70 (0.47 to 1.02). An RCT performed in 18 public schools in Mongolia randomized 8851 schoolchildren aged 6 to 13 years to either 14,000 IU vitamin D weekly or a placebo for 3 years [34]. The primary outcome, i.e., a positive QuantiFERON-TB Gold In-Tube assay test result, was recorded in 3.6% of the children in the vitamin D group and in 3.3% of the children in the placebo group, with a respective HR (95% CI) of 1.10 (0.87 to 1.38; *p* = 0.42). In a trial from Germany, 400 patients with advanced heart failure and 25(OH)D concentrations below 30 ng/mL were randomized to receive either 4000 IU of vitamin D per day or a placebo for 3 years [33]. All-cause mortality was 19.6% in the vitamin D group and 17.9% in the placebo group, with an HR (95% CI) of 1.09 (0.69 to 1.71; *p* = 0.73). In a so-called safety trial from Canada, 311 healthy participants aged 55 to 70 years with a 25(OH)D concentration from 12 to 40 ng/mL were randomized to either 400, 4000, or 10,000 IU of vitamin D per day for 3 years [36]. Primary outcomes were total volumetric bone mineral density (BMD), which was measured by high resolution peripheral quantitative computed tomography, and bone strength (failure load) at the radius and tibia. Compared to the 400 IU group, radial volumetric BMD was significantly lower for the 4000 and 10,000 IU group, and tibial volumetric BMD was significantly lower for the 10,000 IU group, with no significant differences being observed for other primary outcome measures.

### 3.5. Secondary Outcome and Subgroup Analyses

Publications of the above described primary outcomes of the large vitamin D RCTs were accompanied and followed (or in the case of, e.g., the D-Health Trial even preceeded) by a total number of several hundreds of secondary outcome and subgroup analyses (as well as meta-analyses) that should only be interpreted with great caution [35,39,40,41,42,43,44,45,46,47,48]. In general, the risks of type 1 errors (“false positive results”) and type 2 errors (“false negative results”) should be taken into account when interpreting such trial results. Major problems with secondary and subgroup analyses are, e.g., the common lack of pre-specified power analyses, no adjustments for multiple testing (e.g., setting the p value for statistical significance as 0.05 divided by the number of tests according to Bonferroni or using the Benjamini–Hochberg procedure), or missing assumptions of minimal important difference (MID) effect estimates [40]. These limitations, inherent to many RCTs including the large vitamin D RCTs, suggest that the published secondary outcome and subgroup analyses can only be interpreted as so called “explorative outcome” analyses. Therefore, we only briefly discuss some of the findings from such analyses. In general, the vast majority of these explorative outcome analyses did not reveal findings in favour of rejecting the null hypothesis of no vitamin D effect. The indication of no vitamin D effect applied to cardiovascular events, fractures, or falls [26,27]. Of interest were some analyses suggesting a potential beneficial effect of vitamin D on cancer mortality and advanced cancer (metastatic or fatal) [27,38,45]. In this context, vitamin D supplementation significantly reduced cancer mortality in the VITAL study when excluding the first year or the first two years of follow-up [27]. Moreover, in analyes restricted to participants with a body mass index below 25 kg/m^2^, cancer incidence was significantly reduced in the vitamin D when compared to the placebo group [27]. In addition, some explorative analyses suggest that participants with low 25(OH)D concentrations may benefit regarding some surrogate parameters such as BMD, arterial, or lung function [38]. In secondary analyses of the D2d trial, participants who maintained intra-trial 25(OH)D concentrations of at least 40 ng/mL had a significantly reduced risk of progression from prediabetes to diabetes mellitus [49]. By contrast, subgroup analyses from the VIOLET trial in ICU patients with sepsis, infection, or respiratory distress syndrome even suggest increased mortality in patients receiving vitamin D [31]. Similarly, an increased need for mechanical circulatory support (MCS) was observed in the vitamin D group of an explorative analyses of 400 heart failure patients from the EVITA trial [33]. Apart from these two latter studies, there were no major signs of concern regarding adverse clinical events (such as kidney stones) of vitamin D supplementation [46]. With regard to the safety of vitamin D supplementation, it must be emphasized that some previous RCTs using high intermittent doses of vitamin D have partially shown an increased risk of fractures and/or falls, including one RCT documenting an increased risk of falls at a dose of 60,000 IU of vitamin D per month [50,51]. Other RCTs, such as the ViDA study using vitamin D doses of 100,000 IU per month, could, however, not confirm these potential adverse effects of intermittent bolus doses, but uncertainty and concerns remain regarding this issue due to the relatively short half-life of 25(OH)D and other vitamin D metabolites, so it appears prudent to prefer daily or weekly dosing intervals.

## 4. Interpretations and Implications of Large Vitamin D RCTs

As none of the above listed vitamin D RCTs reported any significant effect on the primary outcomes, it has to be concluded that there are no obvious overall major health benefits of vitamin D supplementation in the setting and in the populations included. It can also be concluded that any potential health benefits in the subgroups are likely to be relatively small if present at all. Despite these clear signs of lacking beneficial vitamin D effects, we wish to critically discuss the study designs of these RCTs before drawing final conclusions that may have a huge impact on the current use of vitamin D treatment and future vitamin D research.

It has to be stressed that vitamin D RCTs or nutrient RCTs in general have fundamental differences compared to drug RCTs [52]. With reference to vitamin D, it is not biologically possible that there is no vitamin D exposure in the placebo group, so any group comparison within vitamin D RCTs is always based on a placebo group with a certain vitamin D exposure compared to the intervention group with a higher vitamin D exposure. Therefore, any conclusion that vitamin D supplementation at a given dose is of no health benefit has to refer to the baseline 25(OH)D concentration of the study population and to the achieved vitamin D status after treatment. We therefore outline the 25(OH)D concentrations at baseline and follow-up of important vitamin D RCTs in Table 1 along with the vitamin D dosages and dosing schedules.

The baseline and follow-up vitamin D status of the above-mentioned RCTs has to be viewed in light of observational data on 25(OH)D and hard clinical outcomes such as mortality [21,22,23]. Epidemiological studies from Europe and the US observed the lowest risk of mortality at 25(OH)D concentrations of 31.3 and 32.5 ng/mL, respectively [21,22]. Importantly, data from the Third National Health and Nutrition Survey (NHANES III) showed no association of 25(OH)D with total mortality for values ranging from 16 to 48 ng/mL, but reported a significant increase in mortality at 25(OH)D concentrations below 16 ng/mL [22]. Similar results were observed for associations between 25(OH)D concentrations and incidence rates of cardiovascular disease or cancer [53,54,55]. It is thus in line with (or one may argue a confirmation of) previous observational studies that there was no vitamin D effect on the primary outcomes when studying populations with baseline 25(OH)D concentrations that were mostly above, e.g., 16 ng/mL [56,57]. Bolland and colleagues therefore concluded that these large vitamin D RCTs could be considered research waste because they enrolled participants who were not vitamin D deficient [57]. We are not that critical, as we greatly appreciate the efforts to conduct these large vitamin D RCTs, but we stress that most large vitamin D RCTs enrolled participants in whom, based on their baseline 25(OH)D concentration, no significant effect on the primary outcome could realistically be expected.

Only the VIOLET, EVITA, and MDIG trial enrolled participants with very low 25(OH)D concentrations, but these trials also failed to document significant effects on the primary outcomes [30,31,33]. One conclusion from the VIOLET and EVITA trial in critically ill or severely ill (advanced heart failure) patients is that for such patients, there may be safety concerns when using high dose vitamin D treatment, so we should refrain from this in clinical routine [31,33]. With reference to pregnancy outcomes in the MDIG trial, it has to be acknowledged vitamin D supplementation started relatively late during pregnancy and that the overall rate of pregnancy complications was relatively low for Bangladesh [30]. Moreover, other RCTs and meta-analyses do suggest potential benefits of vitamin D supplementation during pregnancy for e.g. gestational diabetes mellitus or pre-eclampsia [58,59]. In view of the totality of evidence we continue to strongly recommend a sufficient vitamin D status with a 25(OH)D concentration of at least 20 ng/mL in pregnant women. This can be safely and effectively achieved by vitamin D supplementation with a dose of 800 to 1000 IU per day [59,60].

## 5. Meta-Analyses of Vitamin D RCTs

It is also crucial to outline the knowledge provided by meta-analyses of RCTs that have partially considered the evidence provided by the above-mentioned trials [10,11,61,62,63,64,65,66,67].

Regarding cancer outcomes, meta-analyses of RCTs conclude that vitamin D supplementation does not have an effect on cancer incidence, but vitamin D supplementation reduces cancer mortality up to 16% with a respective HR (95% CI) of 0.84 (0.74 to 0.95) based on 12 RCTs in 45,578 participants with 939 cancer deaths [10,11,14,68]. These data on vitamin D and cancer mortality are in line with experimental and observational studies on this topic.

Meta-analyses of RCTs reported that vitamin D supplementation does not reduce overall or individual (such as stroke or myocardial infarction) cardiovascular events [14,26,66]. Similarly, there were also no overall major effects on vascular function or cardiovascular risk factors, although some explorative subgroup analyses of individual RCTs and meta-analyses suggest potential (minor) benefits in subgroups such as those with low 25(OH)D and prediabetes [10,24,29,38,67,69].

With reference to bone health or musculoskeletal health in general, it has to be stressed that the effects of vitamin D in terms of the prevention and treatment of nutritional rickets and osteomalacia are historically established and provide the basis for the conclusion of nutritional vitamin D guidelines that serum 25(OH)D concentrations below 10 or 12 ng/mL have to be prevented and treated [1]. This consensus has not been challenged by recent large vitamin D trials or their meta-analyses. Of note, explorative data from the MDIG trial reported three radiographically confirmed cases of rickets in the placebo group and just one case in the intervention groups that included four times more patients than the placebo group did overall [30]. Regarding other musculoskeletal health outcomes, i.e., fractures, falls, and BMD, the conclusions from meta-analyses of the RCTs are inconsistent and puzzling [10,14,63,64,65]. While some meta-analyses conclude that vitamin D supplementation per se does not prevent fractures and falls or has meaningful effects on BMD, there are other meta-analyses documenting fracture prevention by daily combined calcium plus vitamin D supplementation in older adults [10,14,63,64,65]. Data interpretation has to consider evidence for a U-shaped effect, as high vitamin D bolus doses increase the risk of fractures and falls, whereas any beneficial effects, if present, are particularly observed with moderate vitamin D doses of about 800 IU per day in older individuals with a poor vitamin D status [10,14,63,64,65].

A recently published updated meta-analysis of RCTs reported that vitamin D supplementation significantly prevents acute respiratory infections [62]. By using data from 48,488 participants from 43 RCTs, the odds ratio (95% CI) for having one or more acute respiratory infections in the vitamin D versus the placebo group was 0.92 (0.86 to 0.99). Explorative subgroup analyses indicated a protective effect in groups with daily vitamin D doses, a dose equivalent of 400 to 1000 IU per day, a trial duration of up to 12 months and in those aged 1 to 16 years old. Accordingly, there are also findings from meta-analyses of RCTs suggesting that vitamin D may prevent exacerbations of COPD and asthma [70,71,72].

Coronavirus Disease 2019 (COVID-19), a disease caused by the severe acute respiratory syndrome coronavirus 2 (SARS-CoV-2), has not been addressed in the above listed large vitamin D RCTs or meta-analyses of RCTs [73,74,75]. Although the data on vitamin D and acute respiratory infections are promising and support recommendations for a sufficient vitamin D status during this pandemic, it is unclear whether this also applies to COVID-19 [76,77]. In view of the exploding publication output on vitamin D and COVID-19, we should keep in mind that research findings are less likely to be true in a hot scientific research field (with more scientific teams involved) [16]. This can be partially attributed to the fact that many groups have started to work on this topic, and those with significant findings are more likely to publish their results (publication bias), do not consider the multiple testing problem of many similar investigations around the world and/or are less critical, careful, and balanced when following a publication hype. We should also consider that vitamin D is effective as a preventive measure and not as a therapy for acute respiratory infections using high doses in patients already suffering from severe infection [61,62]. In this context, the subgroup analyses of the VIOLET trial reported increased mortality in those receiving high doses of vitamin D with sepsis, infection, and respiratory distress syndrome [31]. In general, public health strategies to fight the COVID-19 pandemic should primarily consider well established and relevant data, such as the effectiveness of vaccination and protection after natural infection, as well as the enormous age-dependent association of the infection fatality rate [73,74,75].

Older meta-analyses of vitamin D supplementation and all-cause mortality have reported that vitamin D may moderately but significantly reduce all-cause mortality, whereas updated meta-analyses on this topic failed to document a significant effect [66,78,79]. In detail, the relative risks (95% CI) for the effect of vitamin D in two meta-analyses of RCTs were 0.97 (0.93 to 1.02) and 0.98 (0.95 to 1.02), respectively [66,79].

In addition to classic RCTs, we wish to point out the clinical value of Mendelian Randomization (MR) studies that can be considered as a sort of randomized trials of human nature [6,7,14,61,80,81,82]. MR studies use genetically determined 25(OH)D concentrations as an instrumental variable to evaluate associations with clinical outcomes [82]. This strategy, although observational by definition with all inherent limitations, is useful to address the question of causality, as it can be assumed that a certain genetic variant that predisposes people to higher or lower 25(OH)D concentrations should not be associated with common confounding factors [82]. It is beyond the scope of this article to summarize all MR studies on vitamin D, but it bears mentioning that a recent large MR study including more than half a million participants reported an increased all-cause mortality risk for 25(OH)D concentrations below 10 ng/mL [81]. When using a finer stratification of participants in the same study, there was an inverse association between genetically determined 25(OH)D concentrations and mortality up to 16 ng/mL [81]. Findings of another non-linear MR study also suggest that the correction of vitamin D deficiency could reduce cardiovascular disease incidence and blood pressure [83].

Epidemiological studies including data on the use of vitamin D supplementation and baseline as well as follow-up 25(OH)D concentrations may also be of value, but they require careful interpretation in light of their limitations due to their observational study design [84,85,86,87].

## 6. Future Outlook

Results of a few other large vitamin D RCTs will be published in the near future, but given that, e.g., in the D-HEALTH trial, the 25(OH)D concentration in the placebo group is even above 30 ng/mL, we do not expect results that significantly differ from previous studies with similar designs [44].

Considering that even one year after finishing a trial with 20,000 IU of vitamin D per week for 3 to 5 years, there was still a significant difference in serum 25(OH)D concentrations between the vitamin D and the placebo group (i.e., 33.8 versus 29.2 ng/mL), suggesting a much longer half-life of 25(OH)D than the frequently quoted approximately 3 weeks, we recommend evaluating health outcomes in large vitamin D RCTs after finishing the active treatment periods to capture the potential latency or legacy effects of vitamin D [88]. Such an approach has already been used for the EVITA trial, confirming the null effect of vitamin D on total mortality in this cohort of heart failure patients [89].

When designing and interpreting vitamin D RCTs, it has to be considered that there are significant differences regarding the individual molecular responses to vitamin D [90,91]. A variety of nutrients interact with vitamin D and its metabolism, such as, e.g., magnesium, calcium, vitamin K and A, etc. and may thus modulate the individual sensitivity to effects of vitamin D supplementation [92,93]. Similarly, genetic polymorphisms related to VDR signaling or vitamin D metabolism may also contribute to individual differences in the response to vitamin D supplementation [94]. Overall, future vitamin D research should put more emphasis on a personalized approach. The “fire and forget” concept of recent large vitamin D RCTs with a single vitamin D dose that must fit everyone should be modified for future trial designs by enrolling participants who are most likely to benefit from vitamin D treatment and in whom individual differences are accounted for. It should, for example, be recognized that there are higher vitamin D requirements in obese individuals, and pre-specified optimal 25(OH)D concentrations should be targeted by vitamin D dosing adaptations during the trial. Individual participant baseline and achieved 25(OH)D concentrations should therefore be considered for the design and analysis of RCTs. Potential ethnic differences should also be accounted for [95]. Accurate and standardized measurements of vitamin D status are also crucial for future vitamin D trials, and additional measurements of vitamin D metabolites and the consideration of bioavailable fractions are also worth considering [96]. Finally, seasonal variations in vitamin D status and the various different sources of vitamin D should also be considered in the design of future vitamin D trials as well as in the optimal follow-up time for the given outcomes of interest.

## 7. Conclusions

In conclusion, recent RCTs have failed to document significant extra-skeletal benefits of vitamin D supplementation in individuals with largely normal 25(OH)D concentrations. While the limitations of trial designs increased the likelihood of not achieving significant effects on the primary outcomes, we aimed to interpret these findings regarding their potential impact on the current view of vitamin D in terms of clinical practice and science.

Current nutritional vitamin D guidelines are mainly based on the beneficial effects of vitamin D with reference to rickets and osteomalacia [97,98,99,100]. As noted above, this knowledge has not been challenged by recent large vitamin D RCTs, so their null findings do not have an impact on current vitamin D intake recommendations. With reference to other musculoskeletal health outcomes such as fractures and falls, we conclude that increasing vitamin D doses beyond intakes of about 800 to 1000 IU per day does not confer additional benefits but may even be harmful when very high doses are used, particularly intermittent bolus doses [35,36,50,51].

Regarding non-skeletal health outcomes, there is evidence from RCTs suggesting that vitamin D supplementation may prevent acute respiratory infections and cancer mortality. In addition, limited evidence suggests the potential benefits of vitamin D for some other health outcomes such as diabetes mellitus, but this still requires further evaluation. Moreover, vitamin D RCTs indicate that “more is not always better”. Several lines of evidence suggest that moderate daily vitamin D doses are superior to very high daily doses or intermittent high doses [33,35,36,50,51,62]. Nevertheless, vitamin D requirements and the “optimal vitamin D status” may be different according to the outcomes studied, and we do have firm evidence for some kind of threshold effect in the sense that those with very low 25(OH)D concentrations are most likely to benefit from vitamin D treatment [12,21,22,81].

Finally, we want to point out that further large trials using more personalized approaches are needed to evaluate potential effects of vitamin D treatment in populations with severe vitamin D deficiency or certain characteristics that suggest a high sensitivity to vitamin D treatment. From the public health perspective, there is still an urgent need to erase the worldwide burden of vitamin D deficiency with 25(OH)D concentrations below 10 to 12 ng/mL. This requires, beyond a healthy lifestyle that considers nutrition and physical activity, general actions that can be taken to improve vitamin D status such as vitamin D food fortification and vitamin D supplementation in those with vitamin D deficiency [14,101].

## Figures and Tables

**Table 1 nutrients-14-00303-t001:** Baseline and follow-up 25(OH)D concentrations and vitamin D dosing regimens of selected recent large vitamin D RCTs.

Study Acronym or First Author	Study Population	Baseline 25(OH)D in the Entire Cohort (ng/mL)	Baseline 25(OH)D in the Placebo Group (ng/mL)	Follow-Up 25(OH)D in the Placebo Group (ng/mL)	Baseline 25(OH)D in the Vitamin D Group (ng/mL)	Follow-Up 25(OH)D in the Vitamin D Group (ng/mL)	Vitamin D Supplement Dose	Study Duration
VITAL	Older general population	30.8 ± 10.0	30.8 ± 10.0	minus 0.7 from baseline	30.9 ± 10.0	41.8 (mean)	2000 IU per day	5.3 years (median)
ViDA	Older general population	25.3 ± 9.5	24.4 ± 9.6	26.4 ± 11.6	24.4 ± 9.6	54.1 ± 16.0	Initial 200,000 IU, followed by 100,000 IU per month	3.3 years (median)
DO-HEALTH	Older general population	22.4 ± 8.4	22.4 ± 8.5	24.4 (mean)	22.4 ± 8.4	37.6 (mean)	2000 IU per day	2.99 years (median)
D2d	Patients with prediabetes	28.0 ± 10.2	28.2 ± 10.1	28.8 (mean)	27.7 ± 10.2	54.3 (mean)	4000 IU per day	2.5 years (median)
MDIG	Pregnant women	11.0 ± 5.7	11.1 ± 5.5	9.5 ± 5.6	11.0 ± 5.7, 11.5 ± 5.6, 10.8 ± 5.9	27.9 ± 7.8, 40.4 ± 9.4, 44.3 ± 11.2	4200 IU per week, 16,000 IU per week, or 28,000 IU per week	From 17 to 24 weeks of gestation until birth
VIOLET	Critically ill patients	Not reported	11.0 ± 4.7	11.4 ± 5.6	11.2 ± 4.8	46.9 ± 23.2	Single enteral dose of 540,000 IU	90 days
CAPS	Postmenopausal women	32.8 ± 10.5	32.7 (95% CI: 32.1 to 33.3)	30.9 (95% CI: 30.2 to 31.6)	33.0 (95% CI: 32.3 to 33.6)	42.5 (95% CI: 41.7 to 43.3)	2000 IU plus 1500 mg calcium per day	4 years
Ganmaa	School children	11.9 ± 4.2	11.9 ± 4.2	10.7 ± 5.3	11.9 ± 4.2	31.0 ± 9.1	14,000 IU per week	3 years (median)
EVITA	Patients with heart failure	14.6 ± 6.7	14.1 (10.3 to 19.7)	16.3 (12.5 to 23.2)	12.5 (8.6 to 17.9)	37.2 (25.0 to 51.4)	4000 IU per day	3 years
Burt	Older general population	31.3 ± 7.8	No placebo group	No placebo group	30.6 ± 8.4, 32.5 ± 8.0, 31.3 ± 7.4	31.0 (mean), 52.9 (mean), 57.8 (mean)	400 IU per day, 4000 IU per day, 10,000 IU per day	3 years

Data are shown as mean ± standard deviation (SD) or as medians with 25th to 75th percentile, if not otherwise indicated; for the MDIG trial, only groups with no postpartum intervention are shown.

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
