# Peer review of "Critical Appraisal of Large Vitamin D Randomized Controlled Trials"

_nutrients, 2022, doi:10.3390/nu14020303_

Round 1

Reviewer 1 Report

This manuscript outlines the shortcomings of vitamin D RCTs conducted to date. The authors might try to suggest additional ways to revise vitamin D RCTs or alternative approaches to obtain the information not yet obtained from such RCTs.

The main issues regarding vitamin D RCTs:

Half life of 25OHD is about 2 weeks, so monthly supplementation is too infrequent

Doses too small

Vitamin D response

Should measure 25OHD

Too few participants with low 25OHD

Too much other sources of vitamin D

Seasonal variations

Wrong goal in design

Large expense, long duration of RCTs

VITAL is the first large-scale randomized trial of marine omega-3s in a general population at “usual risk” of CVD. It’s also one of the first randomized trials of these supplements in a racially and ethnically diverse study population. Assessing the role of these supplements in a general population free of cardiovascular disease and cancer at baseline fills an important gap in knowledge. Nov. 27, 2018

https://news.harvard.edu/gazette/story/2018/11/vital-researcher-joann-manson-outlines-findings-on-vitamin-d-omega-3/

Comment: the stated goal of the of the RCT was not a scientific one related to determining the relationship between 25(OH)D and incidence of cancer or CVD. In addition, the NIH limited the vitamin D dose to 2000 IU/d because in 2010, there was concern about adverse effects of higher doses. And, to get 25,000 participants, they relaxed all requirements on baseline 25OHD. (For background, probably not for discussion in the manuscript.)

While caution regarding secondary analyses might be justified in general, there seem to be cases when secondary analyses are quite useful. One example comes from the D2d trial in which an analysis of individual participant intratrial analyses of serum 25OHD concentration in the vitamin D treatment arm found significantly reduced risk of progression from prediabetes to diabetes:
Intratrial Exposure to Vitamin D and New-Onset Diabetes Among Adults With Prediabetes: A Secondary Analysis From the Vitamin D and Type 2 Diabetes (D2d) Study.

Dawson-Hughes B, Staten MA, Knowler WC, Nelson J, Vickery EM, LeBlanc ES, Neff LM, Park J, Pittas AG; D2d Research Group.Diabetes Care. 2020 Dec;43(12):2916-2922. doi: 10.2337/dc20-1765. Epub 2020 Oct 5.

The second RCT in which secondary analyses provided useful information in line with general expectations is from VITAL [Ref. 26, Manson et al.]. Cancer mortality rates were significantly lower in the vitamin D treatment arm than in the placebo arm after the first one or two years of data were omitted. Participants with BMI <25 kg/m2 had a significantly reduced risk of cancer than those with higher 25OHD. It is now realized that higher BMI is associated with higher systemic inflammation, a risk factor for many adverse health outcomes. In the supplement to the main paper it was reported that based on data from those who submitted 25OHD concentrations, the three BMI groups had similar increases in 25OHD, approximately 12 ng/ml. Evidently that increase was insufficient for those with higher BMI. Also, black participants in the vitamin D treatment arm had a nearly significant reduced risk of cancer; they had baseline 25OHD concentrations lower than the mean for the entire set of those who submitted values.

Thus, in line with Heaney’s suggestion, individual participant baseline and achieved 25OHD concentrations should be used as the basis for design and analysis of vitamin D RCTs.

Regarding observational studies, these might be cited as they are based on participants supplementing with vitamin D:

The Effects of Vitamin D Supplementation and 25-Hydroxyvitamin D Levels on the Risk of Myocardial Infarction and Mortality.

Acharya P, Dalia T, Ranka S, Sethi P, Oni OA, Safarova MS, Parashara D, Gupta K, Barua RS.J Endocr Soc. 2021 Jul 15;5(10):bvab124. doi: 10.1210/jendso/bvab124. 

The Association between Serum 25(OH)D Status and Blood Pressure in Participants of a Community-Based Program Taking Vitamin D Supplements.

Mirhosseini N, Vatanparast H, Kimball SM.Nutrients. 2017 Nov 14;9(11):1244. doi: 10.3390/nu9111244.

And a cohort study:

Vitamin D and mortality: meta-analysis of individual participant data from a large consortium of cohort studies from Europe and the United States.

Schöttker B, Jorde R, Peasey A, Thorand B, Jansen EH, Groot Ld, Streppel M, Gardiner J, Ordóñez-Mena JM, Perna L, Wilsgaard T, Rathmann W, Feskens E, Kampman E, Siganos G, Njølstad I, Mathiesen EB, Kubínová R, PajÄ…k A, Topor-Madry R, Tamosiunas A, Hughes M, Kee F, Bobak M, Trichopoulou A, Boffetta P, Brenner H; Consortium on Health and Ageing: Network of Cohorts in Europe and the United States.BMJ. 2014 Jun 17;348:g3656. doi: 10.1136/bmj.g3656.

Mendelian randomization studies are improving with the use of stratified genetically-predicted 25(OH)D concentrations and starting to be accepted as showing the causal nature of vitamin D in reducing risk of disease:

Non-linear Mendelian randomization analyses support a role for vitamin D deficiency in cardiovascular disease risk.

Zhou A, Selvanayagam JB, Hyppönen E.Eur Heart J. 2021 Dec 5:ehab809. doi: 10.1093/eurheartj/ehab809. 

Emerging Risk Factors Collaboration/EPIC-CVD/Vitamin D Studies Collaboration

Estimating dose-response relationships for vitamin D with coronary heart disease, stroke, and all-cause mortality: observational and Mendelian randomisation analyses

Lancet Diabetes Endocrinol. 2021 Dec;9(12):837-846.  doi: 10.1016/S2213-8587(21)00263-1. Epub 2021 Oct 28.

From the abstract: “There remains a great need to evaluate effects of vitamin D treatment in popu- 35
lations with vitamin D deficiency or certain characteristics suggesting a high sensitivity to treat- 36
ment.”

Comment: Given the facts that a decade’s worth of effort in vitamin D RCTs has been largely disappointing, and that large-scale vitamin D RCTs are both very expensive and may take up to a decade to complete, it seems that it is time to consider alternatives to RCTs. For example, observational studies based on participants taking vitamin D supplements with additional information about them measured during the follow-up period, such as some of the references cited above and articles such as these:

Maternal 25(OH)D concentrations 40 ng/mL associated with 60% lower preterm birth risk among general obstetrical patients at an urban medical center.

McDonnell SL, Baggerly KA, Baggerly CA, Aliano JL, French CB, Baggerly LL, Ebeling MD, Rittenberg CS, Goodier CG, Mateus Niño JF, Wineland RJ, Newman RB, Hollis BW, Wagner CL.PLoS One. 2017 Jul 24;12(7):e0180483. doi: 10.1371/journal.pone.0180483. 

Breast cancer risk markedly lower with serum 25-hydroxyvitamin D concentrations ≥60 vs <20 ng/ml (150 vs 50 nmol/L): Pooled analysis of two randomized trials and a prospective cohort.

McDonnell SL, Baggerly CA, French CB, Baggerly LL, Garland CF, Gorham ED, Hollis BW, Trump DL, Lappe JM.PLoS One. 2018 Jun 15;13(6):e0199265. doi: 10.1371/journal.pone.0199265.

Author Response

We thank the reviewer for crefully reading our manuscript and making some valuable suggestions. The reviewer comments and our answers to the comments are listed below.

This manuscript outlines the shortcomings of vitamin D RCTs conducted to date. The authors might try to suggest additional ways to revise vitamin D RCTs or alternative approaches to obtain the information not yet obtained from such RCTs.

The main issues regarding vitamin D RCTs:

Half life of 25OHD is about 2 weeks, so monthly supplementation is too infrequent

Doses too small

Vitamin D response

Should measure 25OHD

Too few participants with low 25OHD

Too much other sources of vitamin D

Seasonal variations

Wrong goal in design

Large expense, long duration of RCTs

Response: We agree with this comment as we have already discussed most of the main issues of vitamin D RCTs as mentioned by the reviewer in our manuscript. In addition and according to this comment we now also stress the importance of considering seasonal variations and the different sources of vitamin D as well as the optimal duration of the given trial outcomes.

VITAL is the first large-scale randomized trial of marine omega-3s in a general population at “usual risk” of CVD. It’s also one of the first randomized trials of these supplements in a racially and ethnically diverse study population. Assessing the role of these supplements in a general population free of cardiovascular disease and cancer at baseline fills an important gap in knowledge. Nov. 27, 2018

https://news.harvard.edu/gazette/story/2018/11/vital-researcher-joann-manson-outlines-findings-on-vitamin-d-omega-3/

Comment: the stated goal of the of the RCT was not a scientific one related to determining the relationship between 25(OH)D and incidence of cancer or CVD. In addition, the NIH limited the vitamin D dose to 2000 IU/d because in 2010, there was concern about adverse effects of higher doses. And, to get 25,000 participants, they relaxed all requirements on baseline 25OHD. (For background, probably not for discussion in the manuscript.)

Response: Thank you for sharing this background information with us. 

While caution regarding secondary analyses might be justified in general, there seem to be cases when secondary analyses are quite useful. One example comes from the D2d trial in which an analysis of individual participant intratrial analyses of serum 25OHD concentration in the vitamin D treatment arm found significantly reduced risk of progression from prediabetes to diabetes:
Intratrial Exposure to Vitamin D and New-Onset Diabetes Among Adults With Prediabetes: A Secondary Analysis From the Vitamin D and Type 2 Diabetes (D2d) Study.

Dawson-Hughes B, Staten MA, Knowler WC, Nelson J, Vickery EM, LeBlanc ES, Neff LM, Park J, Pittas AG; D2d Research Group.Diabetes Care. 2020 Dec;43(12):2916-2922. doi: 10.2337/dc20-1765. Epub 2020 Oct 5.

Response: We thank the reviewer for this valuable comment and now stress that in the D2d trial, participants who maintained intra-trial 25(OH)D concentrations of at least 40 ng/mL had a significantly reduced risk of progression from prediabetes to diabetes mellitus.

The second RCT in which secondary analyses provided useful information in line with general expectations is from VITAL [Ref. 26, Manson et al.]. Cancer mortality rates were significantly lower in the vitamin D treatment arm than in the placebo arm after the first one or two years of data were omitted. Participants with BMI <25 kg/m2 had a significantly reduced risk of cancer than those with higher 25OHD. It is now realized that higher BMI is associated with higher systemic inflammation, a risk factor for many adverse health outcomes. In the supplement to the main paper it was reported that based on data from those who submitted 25OHD concentrations, the three BMI groups had similar increases in 25OHD, approximately 12 ng/ml. Evidently that increase was insufficient for those with higher BMI. Also, black participants in the vitamin D treatment arm had a nearly significant reduced risk of cancer; they had baseline 25OHD concentrations lower than the mean for the entire set of those who submitted values.

Response: According to this comment we now mention that cancer mortality was significantly reduced in this trial in the vitamin D versus the placebo group when the first or the first two years of follow-up were excluded. In addition, we also mention that body mass index (BMI) may modify the effect of vitamin D supplementation with respect to cancer outcomes.

Thus, in line with Heaney’s suggestion, individual participant baseline and achieved 25OHD concentrations should be used as the basis for design and analysis of vitamin D RCTs.

Response: We totally agree with this and we have already mentioned this aspect in the discussion and stressed this point again in the manuscript.

Regarding observational studies, these might be cited as they are based on participants supplementing with vitamin D:

The Effects of Vitamin D Supplementation and 25-Hydroxyvitamin D Levels on the Risk of Myocardial Infarction and Mortality.

Acharya P, Dalia T, Ranka S, Sethi P, Oni OA, Safarova MS, Parashara D, Gupta K, Barua RS.J Endocr Soc. 2021 Jul 15;5(10):bvab124. doi: 10.1210/jendso/bvab124.  

The Association between Serum 25(OH)D Status and Blood Pressure in Participants of a Community-Based Program Taking Vitamin D Supplements.

Mirhosseini N, Vatanparast H, Kimball SM.Nutrients. 2017 Nov 14;9(11):1244. doi: 10.3390/nu9111244.

And a cohort study:

Vitamin D and mortality: meta-analysis of individual participant data from a large consortium of cohort studies from Europe and the United States.

Schöttker B, Jorde R, Peasey A, Thorand B, Jansen EH, Groot Ld, Streppel M, Gardiner J, Ordóñez-Mena JM, Perna L, Wilsgaard T, Rathmann W, Feskens E, Kampman E, Siganos G, Njølstad I, Mathiesen EB, Kubínová R, PajÄ…k A, Topor-Madry R, Tamosiunas A, Hughes M, Kee F, Bobak M, Trichopoulou A, Boffetta P, Brenner H; Consortium on Health and Ageing: Network of Cohorts in Europe and the United States.BMJ. 2014 Jun 17;348:g3656. doi: 10.1136/bmj.g3656.

Mendelian randomization studies are improving with the use of stratified genetically-predicted 25(OH)D concentrations and starting to be accepted as showing the causal nature of vitamin D in reducing risk of disease: Non-linear Mendelian randomization analyses support a role for vitamin D deficiency in cardiovascular disease risk.

Zhou A, Selvanayagam JB, Hyppönen E.Eur Heart J. 2021 Dec 5:ehab809. doi: 10.1093/eurheartj/ehab809. 

Emerging Risk Factors Collaboration/EPIC-CVD/Vitamin D Studies Collaboration

Estimating dose-response relationships for vitamin D with coronary heart disease, stroke, and all-cause mortality: observational and Mendelian randomisation analyses

Lancet Diabetes Endocrinol. 2021 Dec;9(12):837-846.  doi: 10.1016/S2213-8587(21)00263-1. Epub 2021 Oct 28.

Response: According to the reviewer comment we have now referenced all these above mentioned studies in our manuscript.

From the abstract: “There remains a great need to evaluate effects of vitamin D treatment in populations with vitamin D deficiency or certain characteristics suggesting a high sensitivity to treatment.”

Comment: Given the facts that a decade’s worth of effort in vitamin D RCTs has been largely disappointing, and that large-scale vitamin D RCTs are both very expensive and may take up to a decade to complete, it seems that it is time to consider alternatives to RCTs. For example, observational studies based on participants taking vitamin D supplements with additional information about them measured during the follow-up period, such as some of the references cited above and articles such as these:

Maternal 25(OH)D concentrations 40 ng/mL associated with 60% lower preterm birth risk among general obstetrical patients at an urban medical center.

McDonnell SL, Baggerly KA, Baggerly CA, Aliano JL, French CB, Baggerly LL, Ebeling MD, Rittenberg CS, Goodier CG, Mateus Niño JF, Wineland RJ, Newman RB, Hollis BW, Wagner CL.PLoS One. 2017 Jul 24;12(7):e0180483. doi: 10.1371/journal.pone.0180483. 

Breast cancer risk markedly lower with serum 25-hydroxyvitamin D concentrations ≥60 vs <20 ng/ml (150 vs 50 nmol/L): Pooled analysis of two randomized trials and a prospective cohort.

McDonnell SL, Baggerly CA, French CB, Baggerly LL, Garland CF, Gorham ED, Hollis BW, Trump DL, Lappe JM.PLoS One. 2018 Jun 15;13(6):e0199265. doi: 10.1371/journal.pone.0199265.

Response: We agree with the reviewer that such approaches and study designs may also be useful, in particular in light of the enormous costs and efforts for large vitamin D RCTs. Therefore, we referenced and discussed the above mentioned manuscripts and their respective study designs. 

Reviewer 2 Report

This article analyzed the outcomes of recent large randomized controlled clinical trials that aimed to evaluate the effect of vitamin D supplementation on the prevention of cardiovascular disease, cancer, and mortality. The authors noticed that these clinical trials essentially generated unsatisfactory results. Subsequently, the authors pointed out problems in these trials and provided meaningful suggestions for future clinical studies.

Considering that vitamin D supplementation faces similar problems in virtually all fields, this article will help other scientists re-think the strategies for vitamin D supplementation studies in the future.

This review is well written overall. However, a minor suggestion is that a native speaker or a professional service checks the language carefully.

Author Response

We thank the reviewer for this positive comment and re-checked and slightly modified the manuscript with the help of a native speaker.